# Revisiting Bellman Errors for Offline Model Selection

**Joshua P. Zitovsky** *
Department of Biostatistics
UNC-Chapel Hill
Chapel Hill, NC 27599

**Daniel de Marchi**
Department of Biostatistics
UNC-Chapel Hill
Chapel Hill, NC 27599

**Rishabh Agarwal**
Google Research, Brain Team
MILA, Université de Montréal

**Michael R. Kosorok**
Department of Biostatistics
UNC-Chapel Hill
Chapel Hill, NC 27599

## Abstract

It is well-known that the empirical Bellman errors are poor predictors of value function estimation accuracy and policy performance. This has led researchers to abandon offline model selection procedures based on Bellman errors and instead focus on directly estimating the expected return under different policies of interest. The problem with this approach is that it can be very difficult to use an offline dataset generated by one policy to estimate the expected returns of a different policy. In contrast, we argue that Bellman errors can be useful for offline model selection, and that the discouraging results in past literature has been due to estimating and utilizing them incorrectly. We propose a new algorithm, *Supervised Bellman Validation*, that estimates the expected squared Bellman error better than the empirical Bellman errors. We demonstrate the relative merits of our method over competing methods through both theoretical results and empirical results on offline datasets from the Atari benchmark. We hope that our results will challenge current attitudes and spur future research into Bellman errors and their utility in offline model selection.

## 1 Introduction

Traditionally, reinforcement learning (RL) has assumed access to an online environment or high-fidelity simulator with which to perform repeated interactions [36]. In contrast, offline RL [3, 24, 26] focuses on training an agent solely from a fixed dataset of observed transitions. By not requiring any further online interactions, offline RL algorithms can significantly improve real-world applicability of RL to settings such as autonomous driving [41] and healthcare [34], where large-scale active data collection may be expensive or unsafe but large data banks of previously-logged interactions are still available.

There has been a recent surge of RL algorithms that can train an agent offline with varying success [11, 15]. Most such algorithms rely on online environment interactions to tune their own hyperparameters. However, applying offline RL algorithms in real-world settings necessitates tuning their hyperparameters offline as well (i.e. offline model selection). Prevalent offline model selection approaches are typically based on offline policy evaluation (OPE) algorithms that estimate the expected returns under a given policy [29, 37, 42]. Unfortunately, accurately estimating the expected returns under one policy using data generated by a completely different behavioral policy can be very difficult. Other approaches that have been explored have been based on empirical Bellman errors as an approximation for the true Bellman errors [13, 21, 29]. These works have found empirical

---

*correspondence to Josh `<joshz@live.unc.edu>`

36th Conference on Neural Information Processing Systems (NeurIPS 2022).

Bellman errors to be poor predictors of value model accuracy, leading to a current belief that Bellman errors are not useful in offline model selection [10, 14, 29].

Instead, our work argues that empirical Bellman errors have been found unhelpful in model selection because they are poor proxies for the true Bellman errors. We propose a new algorithm, *Supervised Bellman Validation (SBV)*, that uses supervised learning to derive a Bellman error-based validation loss function and provide a better proxy for the true Bellman errors. We argue several theoretical benefits of SBV over competing methods and apply SBV to offline Atari datasets [3, 15] containing high-dimensional image observations. We explore conditions in which approaches based on Bellman errors can work well for model selection, and demonstrate that SBV performs competitively when these conditions are satisfied. Finally, we discuss implications of our results and suggest potential avenues for future work.

## 2   Preliminaries

In *offline reinforcement learning (RL)* [24, 26], we have a static dataset $\mathcal{D} = \{(s, a, r, s')\}$ consisting of $|\mathcal{D}|$ transitions $(s, a, r, s')$ where $r \in \mathbb{R}$ is the immediate reward observed after state $s \in \mathcal{S}$ was observed and action $a \in \mathcal{A}$ was taken, before observing the new state $s' \in \mathcal{S}$. We assume the data comes from a Markov decision process (MDP) $\mathcal{M} = (\mathcal{S}, \mathcal{A}, T, R, d_0, \gamma)$ [31] with state space $\mathcal{S}$, action space $\mathcal{A}$, transition probabilities $T(s'|a, s) = \Pr(S_{t+1} = s'|A_t = a, S_t = s)$, reward function $R(S_t, A_t, S_{t+1}) = R_t$, initial state probabilities $d_0(s) = \Pr(S_0 = s)$, and discount factor $\gamma \in (0, 1)$. We can think of the observed state-action pairs in $\mathcal{D}$ as being sampled from (or being identically distributed as) $P_{SA}^{\mu}(s, a) = d^{\mu}(s)\mu(a|s)$ where $\mu$ is the *behavioral policy* and $d^{\mu}$ is the marginal distribution of states over time points induced by policy $\mu$ and MDP $\mathcal{M}$. Likewise, we can think of the observed transitions $(s, a, r, s')$ in $\mathcal{D}$ as being sampled from (or being identically distributed as) $P_T^{\mu}(s, a, r, s') = d^{\mu}(s)\mu(a|s)T(s'|a, s)$ where $r = R(s, a, s')$. If $\mathcal{D}$ was generated by a single stationary policy, $\mu$ is simply equal to this policy. On the other hand, if $\mathcal{D}$ consists of the intermediate transitions observed from training an online RL algorithm, $\mathcal{D}$ can be thought of as being generated from many different policies and $\mu$ becomes a mixture distribution of those policies.

Define the *action-value function* for policy $\pi$ as $Q^{\pi}(s, a) = \mathbb{E}_{\pi}[\sum_{t=0}^{\infty} \gamma^t R_t | S_0 = s, A_0 = a]$ where $\mathbb{E}_{\pi}$ denotes expectation over MDP $\mathcal{M}$ and policy $\pi$. The optimal policy $\pi^*$ is the policy whose action-value function equals the optimal action-value function $Q^*(s, a) = \max_{\pi} Q^{\pi}(s, a)$. For any real-valued function of state-action pairs $Q$ (oftentimes called a *Q-function*), the *Bellman operator* $\mathcal{B}^*$ satisfies:

$$\mathcal{B}^* Q(s, a) = \mathbb{E}[R_t + \gamma \max_{a' \in \mathcal{A}} Q(S_{t+1}, a') | S_t = s, A_t = a] \tag{1}$$

It can be shown that $\pi^* = \pi_{Q^*}$, where $\pi_Q(s) = \text{argmax}_a Q(s, a)$ is the greedy policy of Q-function $Q$, and that $Q = \mathcal{B}^* Q$ if and only if $Q = Q^*$ [36].

For any Q-function $Q$, density $P$ of state-action pairs and dataset of transitions $\mathcal{S} = \{(s, a, r, s')\}$, let $||Q||_P^2 = \mathbb{E}_{(s,a)\sim P}[Q(s, a)^2]$ and $||Q||_{\mathcal{S}}^2 = |\mathcal{S}|^{-1} \sum_{(s,a)\in\mathcal{S}}[Q(s, a)^2]$. Define the *Bellman backup function* for Q-function $Q$ as $(\mathcal{B}^* Q)(s, a)$ and its *Bellman error function* as $(Q - \mathcal{B}^* Q)(s, a)$ (see Equation 1). Likewise, define the *empirical Bellman backups* for Q-function $Q$ and dataset $\mathcal{S}$ as $(\mathcal{B}_{\mathcal{S}} Q)(s, a) = r + \gamma \max_{a'} Q(s', a')$, $(s, a, r, s') \in \mathcal{S}$ and their *empirical Bellman errors* as $Q(s, a) - (\mathcal{B}_{\mathcal{S}} Q)(s, a)$, $(s, a) \in \mathcal{S}$. Q-learning algorithms update a working model based on its empirical Bellman backups so as to approximate $Q^*$ [28, 40]. Throughout we assume all estimated value functions and policies are fitted by Q-learning algorithms.

## 3   Related Work

In the *offline model selection* problem, we have a set of candidate value models $\mathcal{Q} = \{Q_1, ..., Q_M\}$, or estimators for $Q^*$, and our goal is to choose the "best" among them based on some criterion. Here each candidate in $\mathcal{Q}$ was estimated via a Q-learning algorithm with a different hyperparameter configuration. For example, candidates in $\mathcal{Q}$ could include Q-functions obtained from running a single DQN algorithm and stopping at different numbers of training iterations, or it could include Q-functions obtained from running DQN with different Q-network architectures. The most popular approach for offline model selection is to use an *offline policy evaluation (OPE)* algorithm, which estimates the marginal expectation of returns $J(\pi) = \mathbb{E}_{\pi}[\sum_{t=0}^{\infty} \gamma^t R_t]$ under policies of interest $\pi \in \{\pi_{Q_1}, \pi_{Q_2}, ..., \pi_{Q_M}\}$ [12]. For example, *importance sampling* estimators directly estimate $J(\pi)$ from $\mathcal{D}$ by using importance weights to adjust for the distribution shift [30, 37]. *Fitted Q-evaluation* indirectly estimates $J(\pi)$ by estimating $Q^{\pi}$ using another off-policy RL algorithm [25, 29]. Finally,

*model-based* approaches indirectly estimates $J(\pi)$ by estimating the transition kernel $T(\cdot|a,s)$ for all $(a,s) \in \mathcal{A} \times \mathcal{S}$ using standard density estimation techniques [7, 35, 42].

Existing OPE approaches often have difficulties with accurately estimating $J(\pi)$. In particular, importance sampling usually has prohibitively-large estimation variance, FQE introduces its own hyperparameters that cannot be easily tuned offline and model-based approaches can have great difficulty modelling the entire underlying MDP in complex and high-dimensional settings. For these reasons, all of these methods have achieved highly sub-optimal performance on certain environments and most of them have not even been attempted on environments with pixel-valued states [12]. In contrast to estimating the expected returns associated with the greedy policies implicit in $\mathcal{Q}$, another approach is to assess candidates $Q_m, 1 \leq m \leq M$ via their *mean squared Bellman error (MSBE)*:

$$||Q_m - \mathcal{B}^* Q_m||^2_{P^\mu_{SA}} = \mathbb{E}_{(s,a) \sim P^\mu_{SA}} \left[ (Q_m(s,a) - (\mathcal{B}^* Q_m)(s,a))^2 \right] \qquad (2)$$

We discuss theoretical justifications for using the Bellman error for model selection in section 5. To estimate the true MSBE (Equation 2), previous literature has used the *empirical mean square Bellman error (EMSBE)* [13, 21, 29]:

$$|Q_m - \mathcal{B}_\mathcal{D} Q_m||^2_\mathcal{D} = \frac{1}{|\mathcal{D}|} \sum_{(s,a,r,s') \in \mathcal{D}} \left[ \left( r + \max_{a' \in \mathcal{A}} Q_m(s',a') - Q_m(s,a) \right)^2 \right] \qquad (3)$$

The problem is that the EMSBE (Equation 3) is a **biased** estimator of the true MSBE and its minimizer can be an arbitrarily poor estimator of $Q^*$ even in the asymptotic case (see section 5 for details). It is likely for this reason that previous literature has found this to be ineffective at hyperparameter tuning [13, 21].

The work by Farahmand and Szepesvari [10] is more related to our approach in that they propose supervised learning to approximate the true Bellman errors. There are however two major distinctions of Farahmand and Szepesvari [10] over our work: 1) their algorithm is much more sample inefficient as they require partitioning the data into thirds as well as tight probabilistic upper bounds on the *excess risk* of the estimates of $\mathcal{B}^* Q_m$ [38]; 2) they did not investigate the empirical performance of their algorithm or of the MSBE more generally.

## 4  Supervised Bellman Validation

To understand *Supervised Bellman Validation (SBV)*, consider first the case where $Q^*(s,a)$ is actually **known** for state-action pairs $(s,a) \in \mathcal{D}$ and we wish to select the candidates $Q_m \in \mathcal{Q}$ that best estimate $Q^*$. An obvious criterion in this case would be the *value function estimation mean squared error (MSE)*:

$$||Q^* - Q_m||^2_{P^\mu_{SA}} = \mathbb{E}_{(s,a) \sim P^\mu_{SA}} \left[ (Q^*(s,a) - Q_m(s,a))^2 \right]. \qquad (4)$$

While $P^\mu_{SA}$ is unknown, we can estimate the associated expectation by randomly partition $80\%$ of the trajectories present in $\mathcal{D}$ into a training set $\mathcal{D}_T$ and reserving the remaining $20\%$ of trajectories as a validation set $\mathcal{D}_V$. We can then generate candidates $\mathcal{Q} = \{Q_1, ..., Q_M\}$ by running deep Q-learning algorithms on $\mathcal{D}_T$ with $M$ different hyperparameter configurations, and use $\mathcal{D}_V$ to estimate the value function estimation MSE (Equation 4) as $||Q^* - Q_m||^2_{\mathcal{D}_V}$, for each $1 \leq m \leq M$.

In real-world RL, the targets $Q^*(s,a), (s,a) \in \mathcal{D}$ are not known: this is what separates supervised learning from offline RL. Instead of a criterion based on Equation 4, SBV uses a surrogate criterion based on the MSBE $||Q_m - \mathcal{B}^* Q_m||^2_{P^\mu_{SA}}$ (Equation 2). The relationship between estimation error and Bellman error is discussed more in section 5. Similar to what we would do in supervised learning, SBV creates a training set $\mathcal{D}_T$ and a validation set $\mathcal{D}_V$ by randomly partitioning trajectories from $\mathcal{D}$. SBV then trains $M$ Q-functions $\mathcal{Q} = \{Q_1, ..., Q_M\}$ on $\mathcal{D}_T$. Note that the MSBE actually contains **two** unknown quantities: the population density $P^\mu_{SA}$, and the $M$ Bellman backup functions $\mathcal{B}^* Q_m, 1 \leq m \leq M$. From Equation 1, we see that each $(\mathcal{B}^* Q_m)(s,a)$ is just a conditional expectation. Therefore, the $M$ Bellman backup functions can be estimated by running $M$ regression algorithms on $\mathcal{D}_T$, with the $m$th such algorithm estimating $\mathcal{B}^* Q_m$ by fitting a function $f$ to minimize the *Bellman backup MSE*:

$$||\mathcal{B}_{\mathcal{D}_T} Q_m - f||^2_{\mathcal{D}_T} = \frac{1}{|\mathcal{D}_T|} \sum_{(s,a,r,s') \in \mathcal{D}_T} \left[ \left( r + \gamma \max_{a'} Q_m(s',a') - f(s,a) \right)^2 \right]. \qquad (5)$$

---

**Algorithm 1:** Supervised Bellman Validation (SBV)

---

1  Input: Set of observed transitions $\mathcal{D} = \{(s, a, r, s')\}$, hyperparameter configurations to evaluate $\mathcal{H} = \{H_1, ..., H_m\}$
2  Randomly partition trajectories in $\mathcal{D}$ to training set $\mathcal{D}_T$ and validation set $\mathcal{D}_V$
3  **for** *configuration* $m \in \{1, ..., M\}$ **do**
4       Estimate $Q^*$ as $Q_m$ by running a Q-learning algorithm on $\mathcal{D}_T$ with configuration $H_m$
5       Estimate $\mathcal{B}^*Q_m$ as $\widehat{\mathcal{B}}^*Q_m$ by minimizing $||\mathcal{B}_{\mathcal{D}_T}Q_m - f||^2_{\mathcal{D}_T}$ wrt $f$ (see Equation 5)
6       Estimate the MSBE of $Q_m$ as $||Q_m - \widehat{\mathcal{B}}^*Q_m||^2_{\mathcal{D}_V}$ (see Equation 6).
7  **end**
8  Output $Q_{m^*}$ as our estimate of $Q^*$ where $m^* = \mathrm{argmin}_{1 \le m \le M} ||Q_m - \widehat{\mathcal{B}}^*Q_m||^2_{\mathcal{D}_V}$

---

Denote the fitted models from our regression algorithms as $\widehat{\mathcal{B}}^*Q_1, ..., \widehat{\mathcal{B}}^*Q_M$. The MSBE for each candidate $Q_m \in \mathcal{Q}$ can then be estimated as:

$$||Q_m - \widehat{\mathcal{B}}^*Q_m||^2_{\mathcal{D}_V} = \frac{1}{|\mathcal{D}_V|} \sum_{(s,a) \in \mathcal{D}_V} \left[ \left( Q_m(s,a) - (\widehat{\mathcal{B}}^*Q_m)(s,a) \right)^2 \right]. \qquad (6)$$

Our method is summarized in Algorithm 1. Here a hyperparameter configuration $H_m$ specifies all relevant hyperparameters of the Q-learning algorithm, such as the specific training algorithm (e.g. double DQN [16] or duelling DQN [39]), the Q-network architecture and the number of training iterations.

To instantiate SBV, Algorithm 2 provides a practical implementation of SBV for tuning the number of training iterations used by DQN. Here the *Q-Network* is a neural network Q-function $Q_\theta$ with trainable parameters $\theta$ that is trained by DQN to approximate $Q^*$, and $Q_{\theta_{k+1}}$ denotes the trained Q-Network after $k + 1$ iterations of the DQN algorithm. Moreover, the *Bellman Network* is a neural network model $\mathcal{B}_\phi$ with trainable parameters $\phi$ that is updated by SBV after every DQN training iteration to approximate $\mathcal{B}^*Q_\theta$ for the current Q-Network $Q_\theta$. We let $\mathcal{B}_{\phi_{k+1}}$ denote the Bellman network updated by SBV in order to approximate $\mathcal{B}^*Q_{\theta_{k+1}}$.

While the Q-network hyperparameters can be tuned via SBV, the Bellman network itself has hyperparameters which must be tuned. As the Bellman network is solving a series of supervised learning problems, the relevant hyperparameters can easily be tuned offline by minimizing validation Bellman backup MSE on $\mathcal{D}_V$ across the various sets of targets. The ability of SBV to fully tune its own hyperparameters offline gives it real-world applicability and sets it apart from other approaches like fitted Q-evaluation, whose hyperparameters have traditionally been chosen based on online evaluations or previous literature on the environments of interest [29].

## 5 Theoretical Results

We begin by introducing some novel theoretical results to support our algorithm, the proofs of which can be found in section A of the Appendix. Our first theorem, given below, implies that a candidate value model $Q_m$ with sufficiently small Bellman errors should be sufficiently accurate and have

---

**Algorithm 2:** Applying Early Stopping to DQN with SBV

---

1  Input: Set of observed transitions $\mathcal{D} = \{(s, a, r, s')\}$
2  Randomly partition trajectories in $\mathcal{D}$ to training set $\mathcal{D}_T$ and validation set $\mathcal{D}_V$
3  Initialize deep Q-network $Q_{\theta_0}$ with trainable parameters $\theta_0$ and Bellman network $\mathcal{B}_{\phi_0}$ with trainable parameters $\phi_0$
4  **for** *iteration* $k \in \{0, ..., K - 1\}$ **do**
5       Update $\theta_{k+1}$ from $\theta_k$ by running DQN on $\mathcal{D}_T$ for one iteration
6       Update $\phi_{k+1}$ from $\phi_k$ via gradient descent to minimize $||\mathcal{B}_{\mathcal{D}_T}Q_{\theta_{k+1}} - \mathcal{B}_\phi||^2_{\mathcal{D}_T}$ wrt $\phi$ (see 5)
7       Estimate of the MSBE of $Q_{\theta_{k+1}}$ as $||Q_{\theta_{k+1}} - \mathcal{B}_{\phi_{k+1}}||^2_{\mathcal{D}_V}$ (see 6)
8  **end**
9  Estimate $Q^*$ as $Q_{k^*}$ and $\pi^*$ as $\pi_{Q_{k^*}}$ where $k^* = \mathrm{argmin}_{1 \le k \le K} ||Q_{\theta_k} - \mathcal{B}_{\phi_k}||^2_{\mathcal{D}_V}$

---

a greedy policy that gives sufficiently high return. Moreover, if the Bellman errors $Q_m - \mathcal{B}^* Q_m$ are unknown but are estimated with sufficient accuracy, the estimated Bellman errors will still be informative in lower-bounding the estimation accuracy and policy value.

**Theorem 1.** *Let $\widehat{\mathcal{B}}^* Q_m$ be an estimate of $\mathcal{B}^* Q_m$ and assume that $||Q_m - \widehat{\mathcal{B}}^* Q_m||_\infty \leq \epsilon$ and $||\widehat{\mathcal{B}}^* Q_m - \mathcal{B}^* Q_m||_\infty \leq \delta$. Then $||Q - Q^*||_\infty \leq \frac{1}{1-\gamma}(\epsilon + \delta)$ and $||V^{\pi^*} - V^{\pi_Q}||_\infty \leq \frac{2}{(1-\gamma)^2}(\epsilon + \delta)$ where $V^\pi(s) = \mathbb{E}_{a \sim \pi(\cdot|s)}[Q^\pi(s,a)]$.*

Let $F^\mu$ be the cumulative distribution function (CDF) associated with $P_T^\mu$ and let $F^\mathcal{D}$ be the *empirical distribution function (EDF)* of transitions associated with $\mathcal{D}$ [23]. Under general conditions, $F^\mathcal{D}$ converges to $F^\mu$ as $|\mathcal{D}|$ grows large. To study the bias and asymptotic properties of the EMSBE (Equation 3) and SBV (Algorithm 1), we focus on the setting where $|\mathcal{D}| = \infty$ and $F^\mathcal{D} = F^\mu$. Theorem 2 states that the EMSBE has non-negligible bias and that its minimum need not be close to $Q^*$ even with infinite samples. In contrast, Theorem 3 states that SBV correctly recovers the true MSBE and correctly selects the true optimal action-value function $Q^*$, at least in the asymptotic case. These theorems help explain why SBV performs well empirically, both overall and relative to the EMSBE (see section 6).

**Theorem 2.** *Assume that $F^\mathcal{D} = F^\mu$. Then $argmin_{Q:\mathcal{S} \times \mathcal{A} \to \mathbb{R}} ||Q - \mathcal{B}_\mathcal{D} Q||_\mathcal{D}^2 \neq Q^*$ in general and:*

$$||Q_m - \mathcal{B}_\mathcal{D} Q_m||_\mathcal{D}^2 - ||Q_m - \mathcal{B}^* Q_m||_{P_{SA}^\mu}^2 = \mathbb{E}_{(S_t, A_t) \sim P_{SA}^\mu} \left\{ Var \left[ R_t + \max_{a' \in \mathcal{A}} Q_m(S_{t+1}, a')|S_t, A_t \right] \right\}$$

.

**Theorem 3.** *Assume the following: (1) $F^\mathcal{D} = F^\mu$; (2) $\mathcal{D}_T$ and $\mathcal{D}_V$ partition the trajectories present in $\mathcal{D}$; (3) $\mathcal{B}^* Q_m$ is estimated as $\widehat{\mathcal{B}}^* Q_m = argmin_{f:\mathcal{S} \times \mathcal{A} \to \mathbb{R}} ||\mathcal{B}_{\mathcal{D}_T} Q_m - f||_{\mathcal{D}_T}^2$. Then under mild technical conditions and with probability one, $\widehat{\mathcal{B}}^* Q_m = \mathcal{B}^* Q_m$, $||Q_m - \widehat{\mathcal{B}}^* Q_m||_{\mathcal{D}_V}^2 = ||Q_m - \mathcal{B}^* Q_m||_{P_{SA}^\mu}^2$ and $||Q_m - \widehat{\mathcal{B}}^* Q_m||_{\mathcal{D}_V}^2 = 0$ if and only if $Q_m = Q^*$.*

## 6 Experiments

We evaluated Supervised Bellman Validation (SBV; Algorithm 1) on four offline DQN-Replay datasets [3] associated with the following Atari games [4]: Pong, Breakout, Asterix and Seaquest. Atari environments are widely studied in deep RL and possess high-dimensional states, stochastic nonlinear transition dynamics and long-term consequences from taking particular actions. They also possess discrete action spaces amenable to Q-learning algorithms. These properties makes Atari environments challenging and applicable candidates for benchmarking real-world performance of SBV. Following [3], we uniformly sub-sampled 25% of our DQN-replay datasets to obtain four training datasets with 10M transitions and four associated validation datasets with 2.5M transitions. For each training dataset, we evaluate a Q-network after each training iteration across two deep Q-learning configurations, which we dub DQN (S) and DDQN (L). DQN (S) uses the standard DQN architecture [28] and the Adam optimizer following Agarwal et al. [3]. The DDQN (L) configuration is described in section B of the Appendix, and involves a deeper network architecture and double Q-learning target updates [16]. Each training configuration was run for 50 iterations, resulting in 100 candidate Q-networks to evaluate for each training dataset.

We implemented SBV in a manner similar to Algorithm 2 to improve computational efficiency. Moreover, the Bellman network training configuration was tuned offline so as to minimize validation error across the 400 evaluated Q-functions and four Atari games, as discussed in section 4. To demonstrate the efficacy of SBV, we compare it to EMSBE (Equation 3) and *weighted per-decision importance sampling (WIS)* [30]. Full implementation details of SBV and WSE can be found in section C of the Appendix. Due to the computational demands of implementing Fitted Q-evaluation (FQE) and the inability to tune its hyperparameters offline, we have omitted results for FQE for now. We also do not implement model-based approaches here: accurately estimating the underlying MDP of Atari environments offline would be completely novel and extremely challenging, warranting a paper in and of itself.

In Table 1, we calculate the mean return of the top-5 policies according to each method for each environment, and standardize the mean returns to a $[0,1]$ range. We can see here that SBV consistently performs better than competing methods with respect to top-5 policy returns. EMSBE does not perform as well as SBV, but it still performs better than WIS.

Table 1: Standardized Top-5 Policy Mean Returns. A mean return of 0% implies that the method choose the worst five policies possible on the given dataset (based on online evaluations), while a mean return of 100% implies that the method choose the best five policies possible.

| Method | Pong | Breakout | Asterix | Seaquest |
|---|---|---|---|---|
| WIS [30] | 41% | 40% | 50% | 13% |
| EMSBE (Equation 3) | 77% | 43% | 51% | 44% |
| SBV (Ours) | **94%** | **79%** | **76%** | **65%** |

In Figure 1, we plot learning curves for the best configuration for each environment, and indicate the iteration where training was stopped by each method. We can see that SBV performs significantly better than competing methods on average. Moreover, only SBV performs as well or better than applying no early stopping across all environments.

## 7   Discussion and Future Work

Our theoretical and empirical results challenge the current belief in the RL community that Bellman errors are not helpful for offline model selection. In addition to the Atari results described in section 6, we also plan to apply SBV to offline RL problems in physics and healthcare involving much smaller offline datasets [27, 32] to further demonstrate the real-world applicability and robust performance of our method. SBV only requires learning a univariate regression function, making it 1) more statistically efficient than either offline RL or model-based learning; 2) straightforward to tune SBV's own hyperparameters to the dataset at hand; and 3) less biased and more sample-efficient than using no function approximation whatsoever. These features make SBV a competitive candidate for offline model selection.

Our empirical results on Atari games suggest that Bellman error-based model selection can perform well in real-world settings provided certain conditions are met. For instance, model selection algorithms based on the MSBE will not perform well if they do not accurately estimate the MSBE. For this reason, the EMSBE is an ineffective model selector and SBV performance is dependent on estimating $\mathcal{B}^*Q_m$ with sufficient accuracy for all $1 \leq m \leq M$. In the case of our experiments, tuning the Bellman network to the datasets at hand so as to minimize validation MSE was a critical component in achieving adequate performance with SBV. Model selection algorithms such as SBV that estimate the MSBE also fail if the MSBE itself is a poor model selector. For example, for the MSBE to perform well as a model selector, we require there to be sufficient diversity in the observed state-action pairs (see the proof of Theorem 3 given in section A of the Appendix to better understand the importance of this condition).

Finally, Theorem 1 and empirical investigations detailed in section D of the Appendix suggest that the MSBE provides a tight lower bound on estimation error and a loose lower bound on policy value. Consequently, the lower the MSBE value, the more informative it will be in terms of policy

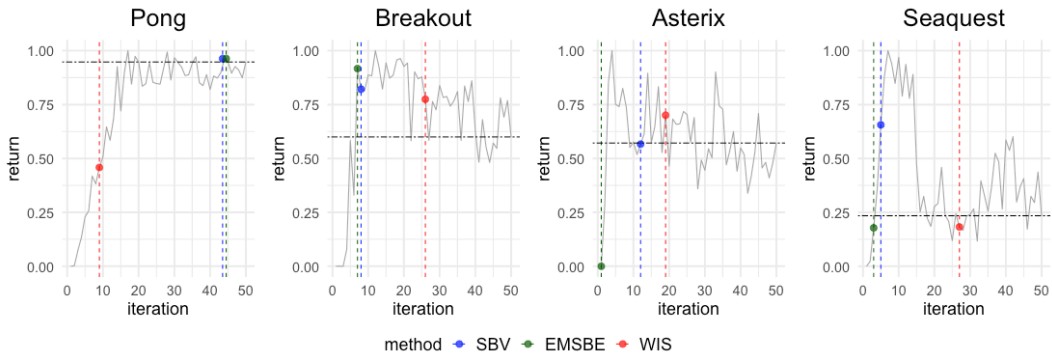

Figure 1: Learning Curves for the Best Configuration. Returns are standardized to a $[0, 1]$ range with zero being the lowest return observed for a given run and one being the highest return observed. The dashed horizontal line represents performance when no early stopping is applied. The vertical lines represent the iterations where training was stopped according to different methods.

performance. For example, if $J(\pi^*) = \mathbb{E}_{\pi^*}[\sum_{t \geq 0} \gamma^t R_t] = 10$, value models with an MSBE of 0.1 may have universally high estimation error but associated policy values in the wide range of $[1, 10]$. On the other hand, value models with an MSBE of only 0.01 would likely have policy values residing in a much tighter range, such as $[9, 10]$ (they will also all have much lower estimation error). The MSBE will thus be an effective model selector when the set of candidate value functions $\mathcal{Q}$ include value functions that have sufficiently small Bellman errors. In real-world settings, we can ensure that our candidate set $\mathcal{Q}$ includes estimates with sufficiently small MSBE by exploring a sufficiently large number of training configurations and/or by exploring training configurations which have been shown in previous work to achieve robust performance in a variety of scenarios.

Our work opens up several avenues for future research. For example, one avenue for future work is to investigate how to achieve sufficient performance with SBV while reducing computational demands. Reducing the number of configurations we have to explore before achieving sufficient performance with SBV would be helpful here, as would modifying the Bellman network training configuration to improve computational efficiency. Moreover, our results focus on discrete control problems and Q-learning algorithms. In contrast, continuous control problems and deep actor-critic algorithms require two neural network models: a Q-network $Q_\theta$ and a *policy model* $\pi_\alpha$. suggests that a $\pi_\alpha \approx \pi^*$, the $\gamma$-contraction properties of the Bellman operator and policy iteration theory [5] suggests that we would need to ensure that both $Q_\theta \approx \mathcal{B}^{\pi_\alpha} Q_\theta$ and that $\pi_\alpha \approx \pi_{Q_\theta}$. While SBV could help ensure that the first condition is satisfied, extending SBV to deal with the second condition is left for future work. Finally, recall that a major shortcoming of FQE is that it has its own hyperparameters that cannot be easily tuned offline. in theory, one can use SBV to tune the hyperparameters of FQE and then use FQE to tune the hyperparameters of deep Q-learning or deep actor-critic. The main issue here is the computational demand: both SBV and FQE are already computationally demanding procedures in their own right, and combining them in a computationally feasible manner is nontrivial.

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

# Appendix

## A   Mathematical Proofs

**Theorem 1.** *Let $\widehat{\mathcal{B}}^* Q_m$ be an estimate of $\mathcal{B}^* Q_m$ and assume that $||Q_m - \widehat{\mathcal{B}}^* Q_m||_\infty \leq \epsilon$ and $||\widehat{\mathcal{B}}^* Q_m - \mathcal{B}^* Q_m||_\infty \leq \delta$. Then $||Q - Q^*||_\infty \leq \frac{1}{1-\gamma}(\epsilon + \delta)$ and $||V^{\pi^*} - V^{\pi_Q}||_\infty \leq \frac{2}{(1-\gamma)^2}(\epsilon + \delta)$ where $V^\pi(s) = \mathbb{E}_{a \sim \pi(\cdot|s)} Q^\pi(s, a)$.*

*Proof.* Suppose $||Q_m - \widehat{\mathcal{B}}^* Q_m||_\infty \leq \epsilon$ and $||\widehat{\mathcal{B}}^* Q_m - \mathcal{B}^* Q_m||_\infty \leq \delta$. Then:

$$\begin{aligned}
||Q_m - Q^*||_\infty &= ||Q_m - \mathcal{B}^* Q_m + \mathcal{B}^* Q_m - Q^*||_\infty \\
&\leq ||Q_m - \mathcal{B}^* Q_m||_\infty + ||\mathcal{B}^* Q_m - Q^*||_\infty && \text{(Subadditivity of } L_\infty \text{ norm)} \\
&= ||Q_m - \mathcal{B}^* Q_m||_\infty + ||\mathcal{B}^* Q_m - \mathcal{B}^* Q^*||_\infty && (Q^* \text{ is a fixed point of } \mathcal{B}^*) \\
&\leq ||Q_m - \mathcal{B}^* Q_m||_\infty + \gamma ||Q_m - Q^*||_\infty && (\mathcal{B}^* \text{ is a } \gamma\text{-contraction}) \\
\Rightarrow ||Q_m - Q^*||_\infty &\leq \frac{1}{1-\gamma} ||Q_m - \mathcal{B}^* Q_m||_\infty \\
&= \frac{1}{1-\gamma} ||Q_m - \widehat{\mathcal{B}}^* Q_m + \widehat{\mathcal{B}}^* Q_m - \mathcal{B}^* Q_m||_\infty \\
&\leq \frac{1}{1-\gamma} \left( ||Q_m - \widehat{\mathcal{B}}^* Q_m||_\infty + ||\widehat{\mathcal{B}}^* Q_m - \mathcal{B}^* Q_m||_\infty \right) && \text{(Subadditivity of } L_\infty \text{ norm)} \\
&\leq \frac{1}{1-\gamma} (\epsilon + \delta)
\end{aligned}$$

Moreover, it was proven in Lemma 1.11 of [2] that $||V^* - V^{\pi_{Q_m}}||_\infty \leq \frac{2}{1-\gamma} ||Q - Q^*||_\infty$. This concludes the proof. $\qquad\square$

**Theorem 2.** *Assume that $F^{\mathcal{D}} = F^\mu$. Then $\operatorname{argmin}_{Q:\mathcal{S}\times\mathcal{A}\to\mathbb{R}} ||Q - \mathcal{B}_{\mathcal{D}} Q||_{\mathcal{D}}^2 \neq Q^*$ and:*

$$||Q_m - \mathcal{B}_{\mathcal{D}} Q_m||_{\mathcal{D}}^2 - ||Q_m - \mathcal{B}^* Q_m||_{P_{SA}^\mu}^2 = \mathbb{E}_{(S_t, A_t)\sim P_{SA}^\mu} \left\{ Var \left[ R_t + \max_{a'\in\mathcal{A}} Q_m(S_{t+1}, a') | S_t, A_t \right] \right\}.$$

*Proof.* Suppose $F^{\mathcal{D}} = F^\mu$. Then:

$$\begin{aligned}
||\mathcal{B}_{\mathcal{D}} Q_m - Q_m||_{\mathcal{D}}^2 &= \mathbb{E}_{(s,a,r,s')\sim P^\mu} \left[ \left( r + \max_{a'} Q_m(s', a') - Q_m(s, a) \right)^2 \right] \\
&= \mathbb{E}_{(s,a,r,s')\sim P^\mu} \left[ \left( r + \max_{a'} Q_m(s', a') - \mathcal{B}^* Q_m(s, a) + \mathcal{B}^* Q_m(s, a) - Q_m(s, a) \right)^2 \right] \\
&= ||\mathcal{B}^* Q_m - Q_m||_{P^\mu}^2 \\
&\quad + \mathbb{E}_{(s,a,r,s')\sim P^\mu} \left[ \left( r + \max_{a'} Q_m(s', a') - \mathcal{B}^* Q_m(s, a) \right)^2 \right] \\
&\quad + 2\mathbb{E}_{(s,a,r,s')\sim P^\mu} \left[ (\mathcal{B}^* Q_m(s, a) - Q_m(s, a)) \left( r + \max_{a'} Q_m(s', a') - \mathcal{B}^* Q_m(s, a) \right) \right]
\end{aligned}$$

Recalling that $\mathcal{B}^* Q_m(s, a) = \mathbb{E}_{s'\sim T(\cdot|s,a)}[R(s, a, s') + \max_{a'} Q_m(s', a')]$ and $P^\mu(s, a, r, s') = P^\mu(s, a) T(s'|s, a)$:

$$\begin{aligned}
&\mathbb{E}_{(s,a,r,s')\sim P^\mu} \left[ \left( r + \max_{a'} Q_m(s', a') - \mathcal{B}^* Q_m(s, a) \right)^2 \right] \\
&= \mathbb{E}_{(s,a)\sim P^\mu} \left\{ \mathbb{E}_{s'\sim T(\cdot|s,a)} \left[ \left( R(s, a, s') + \max_{a'} Q_m(s', a') - \mathcal{B}^* Q_m(s, a) \right)^2 \right] \right\} \\
&= \mathbb{E}_{(S_t, A_t)\sim P^\mu} \left\{ Var \left[ R_t + \max_{a'} Q_m(S_{t+1}, a') | S_t, A_t \right] \right\}
\end{aligned}$$

Moreover:

$$\mathbb{E}_{(s,a,r,s')\sim P^\mu}\left[\left(\mathcal{B}^*Q_m(s,a) - Q_m(s,a)\right)\left(r + \max_{a'}Q_m(s',a') - \mathcal{B}^*Q_m(s,a)\right)\right]$$

$$=\mathbb{E}_{(s,a)\sim P^\mu}\left\{\mathbb{E}_{s'\sim T(\cdot|s,a)}\left[\left(\mathcal{B}^*Q_m(s,a) - Q_m(s,a)\right)\left(R(s,a,s') + \max_{a'}Q_m(s',a') - \mathcal{B}^*Q_m(s,a)\right)\right]\right\}$$

$$=\mathbb{E}_{(s,a)\sim P^\mu}\left\{\left(\mathcal{B}^*Q_m(s,a) - Q_m(s,a)\right)\mathbb{E}_{s'\sim T(\cdot|s,a)}\left[\left(R(s,a,s') + \max_{a'}Q_m(s',a') - \mathcal{B}^*Q_m(s,a)\right)\right]\right\}$$

$$=\mathbb{E}_{(s,a)\sim P^\mu}\left\{\left(\mathcal{B}^*Q_m(s,a) - Q_m(s,a)\right)\times 0\right\} = 0$$

Therefore:

$$||Q_m - \mathcal{B}_\mathcal{D}Q_m||^2_\mathcal{D} = ||Q_m - \mathcal{B}^*Q_m||^2_{P^\mu} + \mathbb{E}_{(S_t,A_t)\sim P^\mu}\left\{\mathrm{Var}\left[R_t + \max_{a'}Q_m(S_{t+1},a')|S_t,A_t\right]\right\}$$

Finally, we need to show that $\mathrm{argmin}_Q||Q - \mathcal{B}_\mathcal{D}Q||^2_\mathcal{D} \neq 0$ in general. In the proof of Theorem 3, we show that $\mathrm{argmin}_Q||Q - \mathcal{B}^*Q||^2_{P^\mu} = Q^*$ (with probability one) under the condition that the behavioral policy $\mu$ is uniformly bounded away from zero. Unless state transitions are deterministic, $\mathbb{E}_{(S_t,A_t)\sim P^\mu}\left\{\mathrm{Var}\left[R_t + \max_{a'}Q(S_{t+1},a')|S_t,A_t\right]\right\} \neq 0$ and thus it is clear that $\mathrm{argmin}_Q\left[||Q - \mathcal{B}^*Q||^2_{P^\mu} + \mathbb{E}_{P^\mu}\left\{\mathrm{Var}\left[R_t + \max_{a'}Q(S_{t+1},a')|S_t,A_t\right]\right\}\right]$ will not be the same as $\mathrm{argmin}_Q||Q - \mathcal{B}^*Q||^2_{P^\mu}$ in general. Specifically, the former will tend to be biased away from $\mathrm{argmin}_Q||Q - \mathcal{B}^*Q||^2_{P^\mu}$ towards lower-variance Q-functions. This concludes the proof.

$\square$

**Theorem 3.** *Assume the following: (1) $F^\mathcal{D} = F^\mu$; (2) $\mathcal{D}_T$ and $\mathcal{D}_V$ partition the trajectories present in $\mathcal{D}$; (3) $\mathcal{B}^*Q_m$ is estimated as $\widehat{\mathcal{B}}^*Q_m = \mathrm{argmin}_{f:\mathcal{S}\times\mathcal{A}\to\mathbb{R}}||\mathcal{B}_{\mathcal{D}_T}Q_m - f||^2_{\mathcal{D}_T}$. Then under mild technical conditions and with probability one, $\widehat{\mathcal{B}}^*Q_m = \mathcal{B}^*Q_m$, $||Q_m - \widehat{\mathcal{B}}^*Q_m||^2_{\mathcal{D}_V} = ||Q_m - \mathcal{B}^*Q_m||^2_{P^\mu_{SA}}$ and $||Q_m - \widehat{\mathcal{B}}^*Q_m||^2_{\mathcal{D}_V} = 0$ if and only if $Q_m = Q^*$.*

*Proof.* (Sketch) In addition to the assumptions given above, we make the following additional technical assumptions: (T1) $\mathcal{D}$ consists of an infinite number of finite-length trajectories $\tau = (s_0, a_0, r_0, s_1, ...., a_T, r_T, s_{T+1})$ distributed iid as $P^\mu_{\mathrm{Traj}}(\tau) = \mathrm{Pr}(T|s_0, a_0, ..., a_T, s_{T+1})d_0(s_0)\prod_{t=0}^T \mu(a_t|s_t)T(s_{t+1}|a_t, s_t)$ (T2) There exists an $\epsilon > 0$ such that $\mu(a|s) > \epsilon$ for all $(s,a) \in \mathcal{S}\times\mathcal{A}$ (T3) $\mathcal{S}$ is countable and $\mathcal{A}$ is finite

Assumption (T3) is made for mathematical convenience. The proofs for when $\mathcal{S}$ and $\mathcal{A}$ are both infinite or even uncountable are similar, but they would require a few minor technical changes and the use of measure theory. When our state-action space is countable, the equalities $\widehat{\mathcal{B}}^*Q_m = \mathcal{B}^*Q_m$ and $\mathrm{argmin}_Q||Q - \widehat{\mathcal{B}}^*Q||^2_{\mathcal{D}_V} = Q^*$ hold *surely*, i.e. for all state-action pairs $(s,a) \in \mathcal{S}\times\mathcal{A}$. For the uncountable case, there could still exist a set of state-action pairs with measure zero such that these equalities do not hold. However, the probability of observing these state-action pairs would be zero under $P^\pi(s,a) = \pi(a|s)d^\pi(s)$ for any policy $\pi$, and thus these equalities still hold *almost surely*, i.e. with probability one (this difference is mathematically important but practically meaningless).

As $\mathcal{D}_T$ and $\mathcal{D}_V$ are constructed by randomly partitioning the trajectories in $\mathcal{D}$ and $\mathcal{D}$ consists of an infinite number of finite-length trajectories with distribution $P^\mu_{\mathrm{Traj}}$, it is easy to see that $\mathcal{D}_T$ and $\mathcal{D}_V$ would themselves consist of an infinite number of trajectories with the same distribution. It is also easy to see that a population distribution of trajectories $P^\mu_{\mathrm{Traj}}$ induces the population distribution of transitions $P^\mu_T$. Thus as the trajectories in $\mathcal{D}_T$ and $\mathcal{D}_V$ are both distributed as $P^\mu_{\mathrm{Traj}}$, $F^{\mathcal{D}_T} = F^{\mathcal{D}_V} = F^\mu$.

Let $P_Y(\cdot|x)$ be the distribution of targets $R_t + \max_{a'}Q_m(S_{t+1},a')$ when $(S_t, A_t) = x$, or when $S_{t+1} \sim T(\cdot|(S_t, A_t) = x)$. As $F^{\mathcal{D}_T} = F^\mu$, we have:

$$||\mathcal{B}_{\mathcal{D}_T} Q_m - f||^2_{\mathcal{D}_T} = \mathbb{E}_{(s,a,r,s') \sim P^\mu} \left[ \left( r + \max_{a'} Q_m(s', a') - f(s,a) \right)^2 \right]$$

$$= \mathbb{E}_{x \sim P^\mu_{SA}, y \sim P_Y(\cdot|x)} \left[ (y - f(x))^2 \right]$$

This is just a population MSE loss function, and it is well-known that the function $f$ minimizing this expectation satisfies $f(x) = \mathbb{E}_{y \sim P_Y(\cdot|x)}[y]$ except possibly for some set $\mathcal{X}^C$ such that $P^\mu_{SA}(x \in \mathcal{X}^C) = 0$ (see for example [17]). However, as $\mu(a|s) > \epsilon > 0$ for all $(s,a) \in \mathcal{S} \times \mathcal{A}$, it is clear that $P^\mu_{SA}(s,a) > 0$ for all $(s,a) \in \mathcal{S} \times \mathcal{A}$, and as $\mathcal{S} \times \mathcal{A}$ is countable, this means that $Pr(X = x) > 0$ for any $x \in \mathcal{S} \times \mathcal{A}$, Therefore, the only such $\mathcal{X}^C$ would be the empty set and $f(x) = \mathbb{E}[Y|X = x]$ for all $x \in \mathcal{S} \times \mathcal{A}$.

As $\mathbb{E}[Y|X] = \mathbb{E}[R_t + \max_{a'} Q_m(S_{t+1}, a')|S_t, A_t] = \mathcal{B}^* Q_m$, we thus have that $\widehat{\mathcal{B}}^* Q_m = \mathcal{B}^* Q_m$. And as $F^{\mathcal{D}_V} = F^\mu$, $||Q_m - \widehat{\mathcal{B}}^* Q_m||^2_{\mathcal{D}_V} = ||Q_m - \mathcal{B}^* Q_m||^2_{P^\mu_{SA}}$. It remains to show that $\mathrm{argmin}_Q ||Q - \widehat{\mathcal{B}}^* Q||^2_{\mathcal{D}_V} = Q^*$, or that $\mathrm{argmin}_Q ||Q - \mathcal{B}^* Q||^2_{P^\mu} = Q^*$. It is easy to show that for any random variable $X$ with distribution $P$, $\mathbb{E}_{x \sim P}[X^2] = 0$ if and only if $\mathrm{Pr}_{X \sim P}(X = 0) = 1$. Therefore $||Q - \mathcal{B}^* Q||^2_{P^\mu} = 0$ if and only if $(Q - \mathcal{B}^* Q)(s,a) = 0$ for all $(s,a) \in \mathcal{S} \times \mathcal{A}$ except possibly some subset $\mathcal{X}^C$ such that $\mathrm{Pr}_{(s,a) \sim P^\mu}((s,a) \in \mathcal{X}^C) = 0$. However, as discussed before, under our assumptions the only subset $\mathcal{X}^C$ that would apply here is the empty set. Therefore $||Q - \mathcal{B}^* Q||^2_{P^\mu} = 0$ if and only if $(Q - \mathcal{B}^* Q)(s,a) = 0$ for all $(s,a) \in \mathcal{S} \times \mathcal{A}$. As $Q^*$ is the unique fixed point of $\mathcal{B}^*$ in $L_\infty$ space, this means that $||Q - \mathcal{B}^* Q||^2_{P^\mu} = 0$ if and only if $Q = Q^*$. As $||Q - \mathcal{B}^* Q||^2_{P^\mu} \geq 0$ for all Q-functions $Q$, it thus holds that any $\mathrm{argmin}_Q ||Q - \mathcal{B}^* Q||^2_{P^\mu} = Q^*$. This concludes the proof.

$\square$

# B   Implementation Details of Q-Learning for Atari

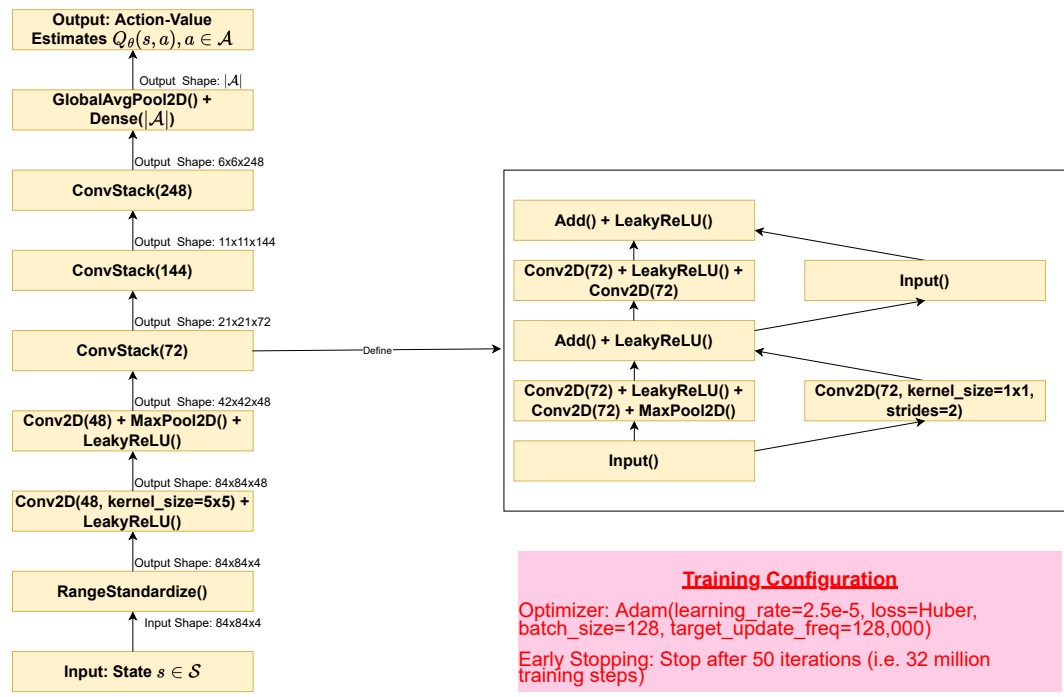

Figure B.1: Network Graph of Q-Network [DDQN (L)]. Unlabeled arrows represent feed-forward connections. Unless otherwise specified, all layers use the default parameters specified by Tensor-Flow [1], with the following exceptions: 1) convolutional layers use 3x3 kernels and zero padding (padding="SAME"); 2) max pooling layers use 3x3 kernels, a vertical and horizontal stride of 2 and apply zero padding (padding="SAME").

There were two training configurations explored when running Q-learning: DQN (S) and DDQN (L). DQN (S) uses the standard Nature DQN architecture and training algorithm [28] and the Adam optimizer following Agarwal et al. [3]. Most of the details of this configuration can be found in Agarwal et al. [3] (they call this configuration "Offline DQN (Adam)") and thus we do not repeat them here. We did however change the batch size from 32 to 128 to speed-up training. We should also mention that our definition of "training iteration" differs from that of Agarwal et al. [3] in that it involves more weight updates to the Q-network overall (640,000 training steps instead of 250,000 training steps) as well as more frequent re-loading of data from disk into memory to reduce the correlation of the mini-batches sampled for training (a new subset of data is loaded into memory every 16,000 training steps instead of every 250,000 training steps). These changes apply to both DQN (S) and DDQN (L).

DDQN (L) uses a deeper and more complex Q-Network architecture that features 16 convolution layers as well max pooling layers, leaky ReLU activation layers [14] and residual connections [18]. The architecture here is simply the best-performing architecture for the Bellman network according to validation Bellman backup MSE that doesn't use batch normalization [20] and is described in more detail below. While there were architectures that performed better as Bellman networks, these all used batch normalization and we found that batch normalization greatly degraded Q-learning performance when added to Q-Networks. We also used double Q-learning targets [16] to reduce overestimation bias. Finally, we tweaked the learning rate and target update frequency so as to improve training stability (though otherwise the optimizer remained unchanged from the DQN (S) configuration). In theory, the learning rate and target update frequency could have been tuned by SBV as well, but to simplify and reduce the computational burden of our experiments, we just fixed these value to 2.5e-5 and 128,000, respectively.

The network graph of the DDQN (L) architecture can be found in Figure B.1. We use Leaky ReLU activation layers in place of the standard ReLU activation layers to fight issues with vanishing gradients and dying ReLUs [14]. A pre-processing layer first scales the input state $s \in \mathcal{S}$ to the $[0, 1]$ range where $\mathcal{S} = \{0, 1, ..., 255\}^{84 \times 84 \times 4}$. After the pre-processing layer, the network starts with two convolutional layers and a pooling layer, with the first convolutional layer having a kernel size of $5 \times 5$ and both convolutional layers having a filter size of $48$. After this is three units which we will call *(DDQN) convolutional stacks*. Each DDQN convolutional stack consists of four convolution layers, all of which have the same number of filters, with a pooling layer inserted after the first two convolution layers. Moreover, the input to the stack is connected to the output of the pooling layer of the stack via a skip connection similar to other ResNet architectures [6], and the input to the third convolution layer is connected to the output of the final convolution layer of the stack via a skip connection as well. The three convolution layers have filter sizes $72$, $144$ and $248$, in ascending order. After the three convolution stacks is a global average pooling layer and a dense layer with units equal to the number of possible actions $|\mathcal{A}|$. The output of this layer gives the Q-network estimates $Q_\theta(s, a), a \in \mathcal{A}$ of the true optimal action-values $Q^*(s, a), a \in \mathcal{A}$ for the given input state $s \in \mathcal{S}$.

## C Implementation Details of SBV and WIS for Atari

### C.1 Bellman Network Architecture, Optimizer and Regularizer

We used two training configurations for the Bellman network: A simpler configuration for Pong and a more complex one for the other three games. The more complex architecture was tuned to minimize validation MSE across the Breakout, Asterix and Seaquest datasets and performs significantly better (in terms of validation MSE) than previously-used architectures and configurations such as Nature [28] and IMPALA [9]. In theory, we could have used a different training configuration for every dataset, or even for every Bellman backup function, but we avoided doing this to simplify the experiments. It is more reminiscent of deep neural networks applied to modern image classification problems such as ImageNet [33] than the small networks employed in deep RL and includes commonly-used design choices in deep learning, such as batch normalization [20] and residual connections [18], to maximize generalization. The simpler configuration applied to Pong was utilized because it achieved almost the same performance as the more complex configuration while requiring significantly less computational resources. The more complex configuration is described below. Details of the simpler configuration are not as important and are not described in this version of the paper.

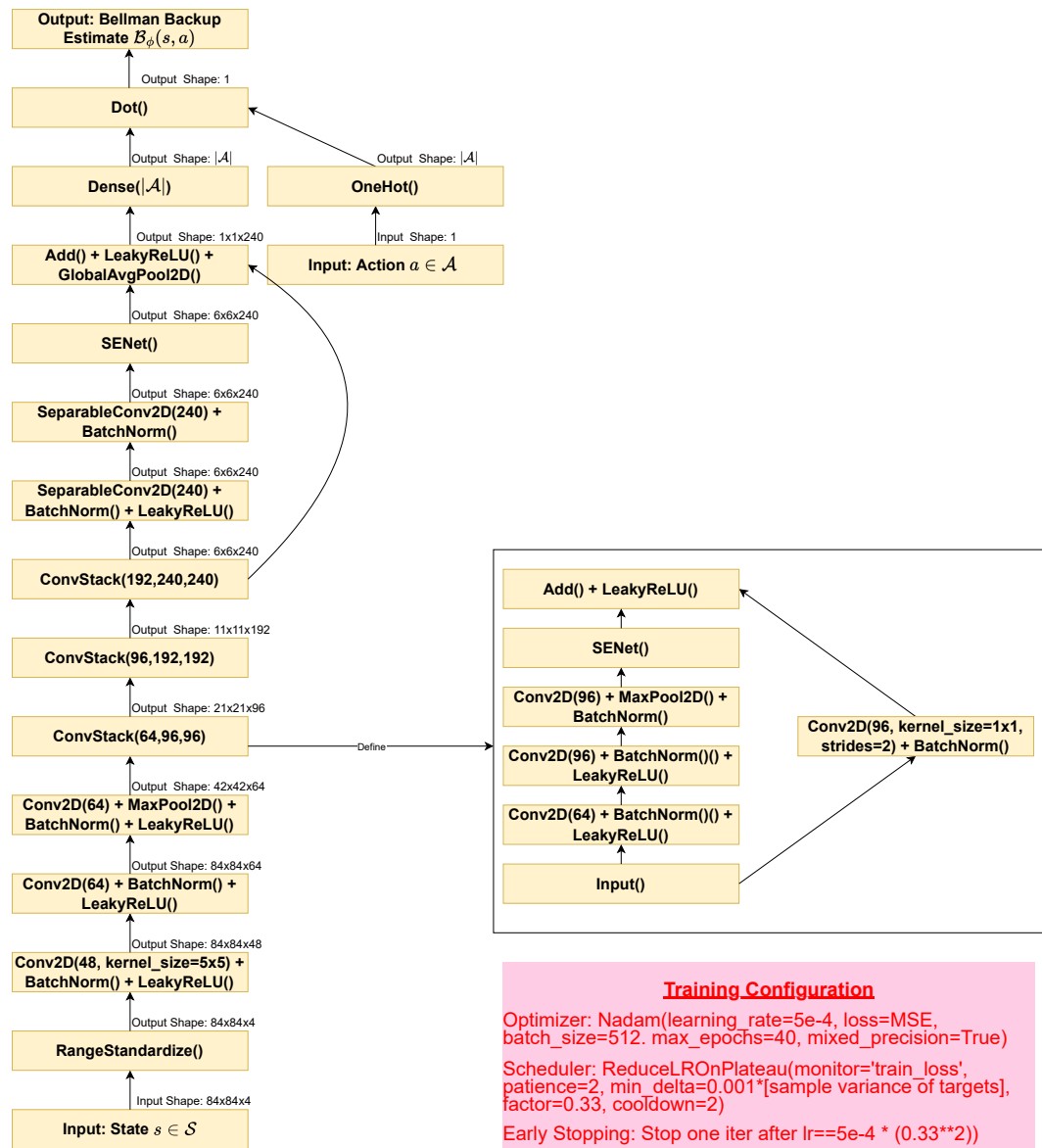

Figure C.1: Network Graph of Bellman Network. Unlabeled arrows represent feed-forward connections. Unless otherwise specified, all layers use the default parameters specified by TensorFlow [1], with the following exceptions: 1) convolutional layers use 3x3 kernels, zero padding (padding="SAME") and no bias; 2) max pooling layers use 3x3 kernels, a vertical and horizontal stride of 2 and apply zero padding (padding="SAME"); 3) SENet() uses an identical architecture to the squeeze-and-excitation units in [19], except the bottleneck layer uses a reduction factor of 4 instead of 16 and a leaky ReLU activation instead of the standard ReLU, and the output layer uses a softplus activation instead of sigmoid.

The network graph can be found in Figure C.1. As is standard in supervised deep convolutional neural networks, We follow every convolutional layer with a batch normalization layer except for those preceding pooling layers: these convolutional layers are followed first by a pooling layer and then by a batch normalization layer. We use Leaky ReLU activation layers in place of the standard ReLU activation layers to fight vanishing gradient and dying ReLU problems [14].

A pre-processing layer first scales the input state $s \in \mathcal{S}$ to the $[0, 1]$ range where $\mathcal{S} = \{0, 1, ..., 255\}^{84 \times 84 \times 4}$. After the pre-processing layer, the network starts with three convolutional layers and a pooling layer. After this is four units which we will call *(Bellman) convolutional stacks*. Each Bellman convolutional stack consists of three convolutional layers, a pooling layer and an SE block [19]. The SE block in this case consists of a global average pooling layer, a hidden dense bottleneck layer with $1/4$th as many hidden neurons as the input width (+ batch norm + leaky ReLU), and an output layer with softplus activation. The input signal is connected to the output of the SE block via a skip connection similar to other SE-ResNet architectures. After the four convolutional stacks are two convolutional layers that use depthwise separable convolutions [6] followed by an SE block. A skip connection connects the input of the first depthwise separable layer to the output of this final SE block.

The number of feature maps per layer increases with depth, increasing from $48$ feature maps for the first convolutional layer to $240$ for the final depthwise separable layers. Following the final SE block is a global average pooling to reduce the number of parameters and a dense layer with units equal to the number of possible actions $|\mathcal{A}|$. The output of the network $\mathcal{B}_\phi(s, a)$ is the dot product between the output of this dense layer and a one-hot transformation of the inputted action $a \in \mathcal{A}$, and is an estimate of the Bellman backup $(\mathcal{B}^*Q_m)(s, a)$ where $Q_m$ is a candidate Q-function that we are trying to evaluate.

The network weights are trained on $\mathcal{D}_T$ to minimize mean square error via the NAdam optimization algorithm [8]. To speed up computations, we enable the mixed precision feature of TensorFlow [1] for the computations and use a batch size of $512$. For the learning rate scheduler, we used an initial learning rate of $5 \times 10^{-4}$ and multiplied the learning rate by a factor of $0.33$ whenever training loss did not improve by at least $0.001 \times \text{var}(\mathcal{B}_\mathcal{D}Q_m(s_t, a_t))$ over $2$ consecutive iterations where $\text{var}(\mathcal{B}_\mathcal{D}Q_m(s_t, a_t))$ is the sample variance of the targets within $\mathcal{D}_T$ being used to train $\mathcal{B}_\phi$. Training was terminated after the first iteration with the learning rate equal to $5 \times 10^{-4} \times 0.33^2$. The performance of the network after each epoch was evaluated by calculating the mean squared error from validation set $\mathcal{D}_V$, and only the weights from the iteration with the lowest validation loss was saved.

## C.2   Other Implementation Details of SBV

Denote $Q_{\theta_k}$ as the trained Q-network after the $k$th iteration for a particular configuration. We wish to estimate the Bellman backup functions $\mathcal{B}^*Q_{\theta_k}, 1 \leq k \leq 50$. Let $\mathcal{B}_{\phi_k}$ denote the trained Bellman network for estimating the $\mathcal{B}^*Q_{\theta_k}$. We perform two additional tricks to speed up computations. First, for most iterations, we initialize the trained weights of $\mathcal{B}_{\phi_{k+1}}$ as $\phi_k$ before conducting training, similar to Algorithm 2 in the main paper. This can be thought of as a kind of transfer learning and greatly speeds up computation. We do, however, occasionally re-randomize initial weights for an iteration via Hu and Xavier initialization schemes to avoid overfitting. The number of consecutive iterations $I$ where weights are transferred from previous iterations before re-randomizing the weights is a hyperparameter that we tuned to minimize validation MSE. In our case, we set $I = 3$ for Seaquest and Breakout datasets, and $I = 1$ for Asterix datasets. The second trick we apply is stopping training once the top-5 policies according to SBV fails to change for five consecutive iterations. By stopping after it is clear that the optimal iteration has been found, we avoid wasting computational resources evaluating Q-functions that aren't going to be selected anyway.

## C.3   Implementation Details of WIS

As WIS requires knowledge of the unknown behavioral policy $\mu$, we estimated it using a neural network approximator $\mu_\beta$ with trainable parameter vector $\beta$ which we dub our *propensity network*. $\mu_\beta$ was trained on $\mathcal{D}_T$ so as to maximize log-likelihood $\sum_{(a,s) \in \mathcal{D}}[\log \mu_\theta(a|s)]$ while its log-likelihood on $\mathcal{D}_V$ was used to tune hyperparameters and evaluate performance. We ended up using the same configuration for the propensity network as that of the Bellman network (Figure C.1), minus the

obvious changes (e.g. MSE loss replaced with categorical cross-entropy loss). We found that this achieved higher validation log-likelihood than other custom configurations explored as well as architectures and configurations used in previous literature such as IMPALA [9].

## D    Supplementary Details on Necessary Conditions for SBV Performance

Theorem 1 and our empirical investigations suggest that for the MSBE to select high-quality policies, we require that our candidate set $\mathcal{Q}$ include estimators of $Q^*$ are sufficiently accurate (i.e. have sufficiently small Bellman errors). To clarify, it is acceptable for $\mathcal{Q}$ to include mostly inaccurate estimates, so long as at least some estimates in $\mathcal{Q}$ are accurate.

To investigate this condition, we tried analyzing only Q-functions generated from DQN (S) by SBV. The result was that SBV failed to properly select high-quality policies for Breakout, Asterix and Seaquest (results not shown). However, once Q-functions generated by DDQN (L) were included in the candidate set, SBV not only correctly identified the Q-functions generated by DDQN (L) as better, but it was also effective at tuning the number of training iterations for DDQN (L), as shown in Table 1 and Figure 1 of the main paper. The issue here is that the Q-network trained by DQN (S) had large Bellman errors at every training iteration despite achieving decent returns from online interactions, while the Q-network trained by DDQN (L) yields much smaller Bellman errors provided we stop training early enough.

It is worth noting however that while $\mathcal{Q}$ needs to include value function estimates with reasonable accuracy, the estimation does not have to be perfect either. For example, state-of-the-art online RL methods have managed to achieve policies with returns of over 100,000 for Seaquest [22]. As the Q-functions generated by DDQN (L) for Seaquest give values of less than 25,000, they must have decent Bellman and estimation error. Despite this, they are good enough for SBV to select high-performing policies.

