# OpenReview forum: "Revisiting Bellman Errors for Offline Model Selection"
_NeurIPS.cc/2022/Workshop/Offline_RL — Offline RL Workshop NeurIPS 2022_

### Official Review · Reviewer_G87s · 2022-10-19
**Supervised variant of MSBE for model selection; not overwhelmingly convincing premise or evaluation, but relevant to workshop topic**

**Rating:** 6
**Confidence:** 4

**Review:**

This paper proposes the supervised Bellman validation (SBV), which involves (1) running a $Q$-learning algorithm to obtain a set of $Q$-functions (ie, from different hyperparameter configurations); (2) fitting a parametric function $f$ to $\hat{B}^* Q $; (3) using the resulting MSBE between the $Q$s and $f$s on a validation set to do model selection. This seems to differ from standard practice only in that a separate parametric function $f$ is fit to $\hat{B}^*Q$ instead of using a conventional exponential moving average of a target network. It is possible that the dense presentation and page limit hid other differences, but I would be somewhat surprised if this difference alone would lead to reliable Bellman errors where they otherwise would not be.

Of course, this is primarily an empirical question at that point. Unfortunately, the empirical evaluation is not particularly convincing. To support the claim that SBV is useful for model selection, I would want to see, eg, a correlation between the proposed metric and another quantity of interest, such as empirical returns.